# The Erasmus+ Programme and Sustainable Development Goals—Contribution of Mobility Actions in Higher Education

Teresa Nogueiro [1], Margarida Saraiva [1,2,*], Fátima Jorge [1,3] and Elisa Chaleta [4,5]

1   Management Department, School of Social Sciences, Universidade de Évora, 7004-516 Évora, Portugal; t.nogueiro@gmail.com (T.N.); mfj@uevora.pt (F.J.)
2   BRU—Business Research Unit-Iscte-Instituto Universitário de Lisboa, 1649-026 Lisboa, Portugal
3   CICP—Research Center in Political Science, Universidade de Évora, 7004-516 Évora, Portugal
4   Psychology Department, School of Social Sciences, Universidade de Évora, 7004-516 Évora, Portugal; mec@uevora.pt
5   CIEP—Center for Research in Education and Psychology, Universidade de Évora, 7004-516 Évora, Portugal
*   Correspondence: msaraiva@uevora.pt

**Abstract:** Erasmus+ is an EU programme in the fields of education, training, youth and sport for the period 2014–2021 with a major impact at the international level. These areas are making important contributions to help address socioeconomic changes and the key challenges that Europe will face until the end of the decade as well as to support the implementation of the European policy agenda for growth, jobs, equity and social inclusion. The general objectives of the Erasmus+ Programme are intended to contribute to the overall achievement of the objectives of the Europe 2020 Strategy, among others related to sustainable development. Therefore, the main question is: "To which Sustainable Development Goals (SDG) do the Erasmus+ Programme and the mobility projects for higher education directly contribute to?" The answer to this question will allow us to start filling the knowledge gap at the borderline between SDG and Erasmus+. The purpose of this research is to identify, among the 17 SDGs, those that could be more relevant in the context of mobility projects in higher education within the Erasmus+ Programme and how these projects contribute to these identified SDGs. Through the analysis of the general objective of the Erasmus+ Programme, its most important characteristics and the objectives of Key Action 1 and mobility projects, we can conclude that, from the 2030 Agenda, the most relevant SDGs for the programme and for the action and mobility projects are 4 (quality education), 5 (gender equality) and 8 (decent work and economic growth).

**Keywords:** Sustainable Development Goals; higher education; Erasmus+ Programme; mobility projects

## 1. Introduction

Scientific papers addressing the issue of Sustainable Development Goals (SDGs) have been multiplying, not because it seems to be fashionable, but because it seems to be of great general importance in this century. The SDGs, which represent and define the global priorities for the 2030 Agenda signed by around 190 countries, seek to mobilise global efforts around a set of common goals and targets. The 17 SDGs are defined in areas that affect the quality of life of all citizens of the world and of future generations. According to the United Nations, the 2030 Agenda and the 17 Sustainable Development Goals are the common vision for humanity, a contract between world leaders and people and "a list of things to do on behalf of people and planet" [1].

The Earth Charter, which emerged from an eight-year consultation of thousands of people from several countries, is considered a document that warns of the risks hanging over humanity but at the same time brings hope by sharing values and principles capable of opening up a new future for everyone on this planet that is Earth [2,3]. According to [2], sustainability was initially seen as a matter of life and death, due to the risks that, at the time of the publication of the Earth Charter, seemed to threaten the future of the planet.





Sustainability is the set of processes and actions that are intended to maintain the vitality and integrity of Mother Earth, the preservation of its ecosystems with all the physical, chemical and ecological elements that enable the existence and reproduction of life, the meeting of the needs of present and future generations and the continuity, expansion and realization of the potential of human civilization in its various expressions [2]. Sustainability is a way of being and living that requires aligning human practices with the limited potential of each biome and the needs of present and future generations.

"Sustainable", "sustainability" and "sustainable development" are terms that have gained relevance at global level, initially associated with planet preservation issues and currently more associated with human well-being and relationship with the planet for the well-being of several generations.

These terms have faced the need to be re-examined, since society has used them ambiguously, confusing them, on certain occasions, with the idea of growth, progress, maturity, evolution or wealth [4]. Ref. [5] emphasizes that sustainability is not just a "fashion or trend" valued by circumstantial conditions, but its importance is linked to the ethics that guide human conduct, reflecting the values of courage, prudence and hope. The term "sustainability" is currently popular, but the concept itself has ancient and universal roots.

According to [6], to be sustainable does not have only one definition and the term "sustainability" expresses the concern with the quality of a system regarding inseparable integration (environmental and human), evaluating properties and characteristics of its own and covering environmental, social and economic aspects. Finally, they refer that the notion of sustainable development presents intrinsic information and should be observed in the development of strategies. Relating the three concepts, [6] (p. 676) states that "Sustainability consists of a goal or parameter (end goal) defined by means of scientific criteria, which measures and monitors the results generated by the use of sustainable development strategies." In the end, these authors further summarize the relationship between sustainability and sustainable development. Table 1 summarizes in brief their considerations.

**Table 1.** Sustainable, sustainable properties and development and the sustainability–sustainable development relationship.

| Sustainable | Sustainability Properties | Sustainable Development | Relationship between Sustainability and Sustainable Development |
|---|---|---|---|
| 1. Solution to natural resource scarcity linked to energy and natural resources issues 2. Originated from the deterioration between global ecology and economic development 3. Covers sustainability and sustainable development 4. Concern for the future of natural resources and human life | 1. Quality and ownership of the global human environmental system 2. Considers the dynamic temporal evolutions 3. Covers environmental, economic and social aspects 4. Mutual balance 5. Evaluation with indicators and indexes | 1. Aims at economic growth without human environmental aggression 2. Long-term vision in relation to future generations 3. Covers the environmental, economic and social aspects in mutual balance 4. Proposes a change in the behaviour of humankind 5. Materialized through strategies 6. Involves processes and practices | Human needs and well-being <==> Global human environmental system Access way <==> Ultimate intent (long-term) Strategies <==> Goal (parameter) Capitalism <==> Ecology Economic <==> Environmental |

Source: Adapted from [6].

Growing awareness of the threat of global warming, the universal understanding of sustainability or sustainable development, in recent times has grown steadily and universally [5].

In society, the social, cultural and psychological dimensions are responsible for changing individual behaviour. Consequently, these are strongly influenced by the political dimension, which is mainly characterised by power games, and the economic dimension. Considering the paradigm of consumption and work imposed by the so-called capitalist society, the spatial dimension is affected. The origin of the ecological dimension is due to the negative impacts that cause the ecosystem to have a worrying imbalance in the maintenance of life on earth, and life on earth is affected by the way of human existence, that is, affected by other dimensions. The dimensions of sustainability are extremely interconnected and it is necessary to find a balance between the proposed dimensions, just as society seeks sustainable development [7].

The world's most pressing issues have been identified, and they range from poverty and hunger elimination to economic growth methods and social necessities, such as education, health, social protection and work possibilities, to climate change and environmental protection [8].

With the establishment of the Millennium Development Goals, developed by the United Nations in 2000, sustainability was defined as one of the primary objectives within the context of globalisation. When it became clear that the desired impact was not being realised in 2015, it was agreed that sustainability should take centre stage, and the Sustainable Development Goals (SDGs) were created [9]. Global development gained a more comprehensive component as a result of this reformulation, which was strongly related to sustainability under the 2030 Agenda [10].

Ban Ki-Moon, eighth Secretary-General of United Nations, in the 159th Grips Forum 2018 stated, "SDGs are by far the most scientific, most comprehensive, most ambitious set of goals that the United Nations has ever presented to the world. The SDGs aim to address the unfinished business of the MDGs. There is a great expansion of scope in the 17 goals, which range from completely eliminating abject poverty to forging global partnerships. Leave no one behind. There are many people who are living in very dire circumstances, including people with physical disabilities, girls and women, people of different sexual orientation, and groups who are excluded by discrimination from normal life in our societies. We have to make absolutely sure that all these people are included in our work to make this a fair world" [11]. The 17 SDGs are in Table 2 as follows:

The Sustainable Development Goals, or SDGs, are a significant move forward in sustainable development, taking a much broader view of sustainability than has ever been attempted before. However, practical obstacles, such as how to implement change, remain [13].

Higher education institutions (HEIs) play a critical role since they are thought and opinion makers, capable of facilitating the development and diffusion of long-lasting ideas. As a result, people involved in the development of university activities must serve as a foundation for knowledge diffusion and the reinforcement of sustainable practices [14]. HEIs are vital to society's development since they are one of the primary actors in the transformation of professionals who will dictate market and societal directions. Because of the vast flow of people, information and activities created and released, HEIs, like any other organization, use a large number of available resources (inputs/outputs). These organizations end up with a significant environmental liability, necessitating the incorporation of sustainable development approaches into their operations [15].

Under this framework, "universities are challenged to include the 17 Sustainable Development Goals (SDG) in the wide range of their training offers and that higher education is expected to contribute knowledge and innovation to meet societal, economic and environmental challenges through the training of both academic staff and students" [8] (p. 2).

**Table 2.** Sustainable Development Goals.

| | | | | | |
|---|---|---|---|---|---|
| 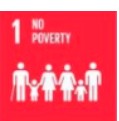 | End poverty in all its forms everywhere | 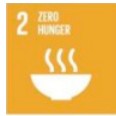 | End hunger, achieve food security and improved nutrition and promote sustainable agriculture | 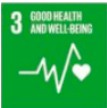 | Ensure healthy lives and promote well-being for all at all ages |
| 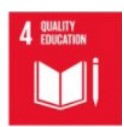 | Ensure inclusive and equitable quality education and promote lifelong learning opportunities for all | 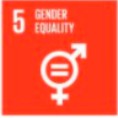 | Achieve gender equality and empower all women and girls | 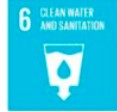 | Ensure availability and sustainable management of water and sanitation for all |
| 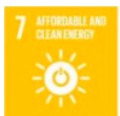 | Ensure access to affordable, reliable, sustainable and modern energy for all | 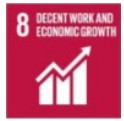 | Promote sustained, inclusive and sustainable economic growth, full and productive employment and decent work for all | 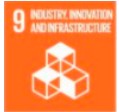 | Build resilient infrastructure, promote inclusive and sustainable industrialization and foster innovation |
| 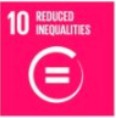 | Reduce inequality within and among countries | 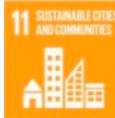 | Make cities and human settlements inclusive, safe, resilient and sustainable | 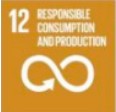 | Ensure sustainable consumption and production patterns |
| 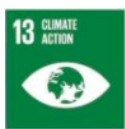 | Take urgent action to combat climate change and its impacts | 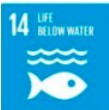 | Conserve and sustainably use the oceans, seas and marine resources for sustainable development | 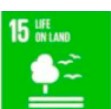 | Protect, restore and promote sustainable use of terrestrial ecosystems, sustainably manage forests, combat desertification and halt and reverse land degradation and halt biodiversity loss |
| 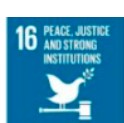 | Promote peaceful and inclusive societies for sustainable development, provide access to justice for all and build effective, accountable and inclusive institutions at all levels | 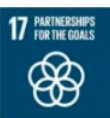 | Strengthen the means of implementation and revitalize the global partnership for sustainable development | 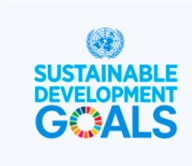 | |

Source: Adapted from [12].

Erasmus+ is the EU's education, training, youth and sport programme for the years 2014–2020. Education, training, youth and sport may all play an important role in addressing socioeconomic changes as well as the significant issues that Europe will face until the end of the decade and in promoting the European policy agenda for growth, jobs, equity and social inclusion. One of the most critical matters for European governments is to reduce high levels of unemployment, particularly among young people. Too many young people drop out of school early, putting themselves at danger of unemployment and social exclusion. Many adults with inadequate skills face the same problem. Technology is redefining the way society works, and it is critical to make the greatest use of it. Through skill and innovation, EU businesses must become more competitive. More cohesive and inclusive societies are needed in Europe, allowing citizens to participate actively in democratic life. To promote common European values, develop social integration, improve intercultural understanding and a sense of belonging to a community and prevent violent radicalization, education, training, youth work and sport are essential. Erasmus+ is a powerful tool for promoting the participation of individuals from underrepresented groups, such as newly arriving migrants. The Erasmus+ Programme is aimed to help Programme Countries' efforts to effectively employ Europe's talent and social assets in a lifelong learning context, combining support for formal, non-formal and informal learning throughout the education,

training and youth sectors. The Programme also expands chances for cooperation and mobility among Partner Countries, particularly in higher education and youth [16].

The general objectives of the Programme are intended to contribute to the overall achievement of the objectives of the Europe 2020 Strategy, among others related to sustainable development. Therefore, the main question was: "To which SDGs are the programme overall and, in particular, the mobility programmes associated with higher education directly contributing?". The answer to this question will allow us to start filling the knowledge gap at the borderline between SDG and Erasmus+. Through the analysis of the general objective of the Erasmus+ Programme, its most important characteristics and the objectives of Key Action 1 and mobility projects, we can conclude that SDGs 4 (quality education), 5 (gender equality) and 8 (decent work and economic growth) are very relevant for the programme and for the action and mobility projects.

## 2. Materials and Methods

For the purpose of this work, the team and analysed the following data, identified from [16]:

1. The objectives of the Erasmus+ Programme and its main features;
2. The specific objectives pursued by the Erasmus+ Programme in the field of education and training;
3. The aims of a mobility project;
4. The outcomes meant to be produced by mobility activities of students, trainees, apprentices and young people;
5. The outcomes expected to be produced by mobility activities of youth workers and professionals involved in education, training and youth.

To analyse that correspondence between the selected aspects in the framework of Erasmus+ and the Sustainable Development Goals, the team analysed the targets, the means of implementation and the respective indicators for each of the 17 SDGs mentioned on [12].

Although the contents of SDGs were analysed individually, it was noticed that there are relationships with variable degrees of strength between them. However, in this work, we will make an individual analysis of each SDG and will present those that seemed more consistent in relation to the contributions that the Erasmus+ Programme could give to its effective implementation.

**Question:** To which SDGs the Erasmus+ Programme and the mobility projects for higher education directly contributing?

## 3. Results

The main result is that Erasmus+ Programme and Key Action 1—mobility of individuals (International Credit Mobility projects included)—clearly contribute to sustainable development goals, namely for SDGs 4, 5 and 8.

### 3.1. Erasmus+ Programme and SDGs 4, 5 and 8

Analysing the targets, means of implementation and indicators of SDGs 4, 5 and 8 (Appendix A), we observe that not all are relevant or related to the Erasmus+ Programme's objectives and important features.

The Erasmus+ Programme has a general objective with which it must contribute to the achievement of:

(1) The objectives of the Europe 2020 Strategy, including the headline education target;
(2) The objectives of the strategic framework for European cooperation in education and training [17], including the corresponding benchmarks;
(3) The sustainable development of Partner Countries in the field of higher education;
(4) The overall objectives of the renewed framework for European cooperation in the youth field;

(5) The objective of developing the European dimension in sport, in particular grassroots sport, in line with the EU work plan for sport;

(6) The promotion of European values in accordance with Article 2 of the Treaty on the European Union.

The important features of the programme that deserve special attention are the recognition and validation of skills and qualifications, the dissemination and exploitation of project results, the Erasmus+ open access requirement for educational materials, the Erasmus+ open access to research and data, the international dimension, multilingualism, equity and inclusion and the protection and safety of participants.

The Erasmus+ Programme's specific goals in the sphere of education and training are as follows:

(a) To improve the level of key competences and skills, with particular regard to their relevance for the labour market and their contribution to a cohesive society, in particular through increased opportunities for learning mobility and through strengthened cooperation between the world of education and training and the world of work;

(b) To foster quality improvements, innovation excellence and internationalisation at the level of education and training institutions, in particular through enhanced transnational cooperation between education and training providers and other stakeholders;

(c) To promote the emergence and raise awareness of a European lifelong learning area designed to complement policy reforms at national level and to support the modernisation of education and training systems, in particular through enhanced policy cooperation, better use of EU transparency and recognition tools and the dissemination of good practices;

(d) To enhance the international dimension of education and training, in particular through cooperation between Programme and Partner Country institutions in the field of VET and in higher education, by increasing the attractiveness of European higher education institutions and supporting the EU's external action, including its development objectives, through the promotion of mobility and cooperation between Programme and Partner Country higher education institutions and targeted capacity building in Partner Countries;

(e) To improve the teaching and learning of languages and promote the EU's broad linguistic diversity and intercultural awareness.

After analysis, the relevant targets and means of implementation detected were for SDG 4: 4.3, 4.4, 4.5, 4.7, 4.b and 4.c; for SDG 5: 5.1 and 5.c and for SDG 8: 8.3 and 8.5.

*3.2. Mobility Projects/International Credit Mobility and SDGs 4, 5 and 8*

Analysing the targets, means of implementation and indicators of SDGs 4, 5 and 8 (Appendix A), we observe that, once again, not all are relevant or related to mobility projects (International Credit Mobility projects included).

The programme defined the following goals for the mobility projects:

(1) To support learners in the acquisition of learning outcomes (knowledge, skills and competences) with a view of improving their personal development, their involvement as considerate and active citizens in society and their employability in the European labour market and beyond;

(2) To support the professional development of those who work in education, training and youth with a view of innovating and improving the quality of teaching, training and youth work across Europe;

(3) To notably enhance the participants' foreign languages competences;

(4) To raise participants' awareness and understanding of other cultures and countries, offering them the opportunity to build networks of international contacts, to actively participate in society and develop a sense of European citizenship and identity;

(5) To increase the capacities, attractiveness and international dimension of organisations active in the education, training and youth fields so that they are able to offer activ-

ities and programmes that better respond to the needs of individuals, within and outside Europe;

(6)    To reinforce synergies and transitions between formal and non-formal education, vocational training, employment and entrepreneurship and to ensure a better recognition of competences gained through learning periods abroad.

Beside the above-mentioned goals, this action also contributes to (1) cooperation between the EU-eligible Partner Countries, reflecting the EU's external action goals, priorities and principles, such as enhancing the attractiveness of higher education in Europe and supporting European higher education institutions in competing on the higher education market worldwide; (2) support of the priorities identified in the "New European Consensus on Development" and the "European Higher Education in the World" Communication; (3) support of the internationalisation, attractiveness quality, equity of access and modernisation of higher education institutions outside Europe in view of promoting the development of Partner Countries and (4) promotion of development and external policy objectives and principles, including national ownership, social cohesion, equity, proper geographical balance and diversity.

Special attention will be given to the least developed countries as well as to disadvantaged students from poor socioeconomic backgrounds and to students with special needs; it will also be given to promoting non-formal learning and cooperation in the field of youth with Partner Countries.

On the framework of mobility projects, several outcomes are expected. For students, trainees, apprentices and young people, the mobility activities are meant to improve learning performance, enhance employability and improve career prospects, increase sense of initiative and entrepreneurship, increase self-empowerment and self-esteem, improve foreign language competences, enhance intercultural awareness, promote more active participation in society, improve awareness of the European project and the EU value; and increase motivation for taking part in future (formal/non-formal) education or training after the mobility period abroad.

For youth workers and professionals involved in education, training and youth, the mobility activities are expected to improve competences, linked to their professional profiles (teaching, training, youth work, etc.); broaden understanding of practices, policies and systems in education, training or youth work across countries; increase capacity to trigger changes in terms of modernisation and international opening within their educational organisations; increase understanding of interconnections between formal and non-formal education, vocational training and the labour market, respectively; improve quality of their work and activities in favour of students, trainees, apprentices, pupils, adult learners, young people and volunteers; deepen understanding and responsiveness to social, linguistic and cultural diversity; increase ability to address the needs of the disadvantaged; increase support for and promotion of mobility activities for learners; increase opportunities for professional and career development; improve foreign language competences and increase motivation and satisfaction in their daily work.

After analysis, the relevant targets and means of implementation detected were for SDG 4: 4.3, 4.4, 4.5, 4.7, 4.b and 4.c; for SDG 5: 5.1 and 5.c and for SDG 8: 8.3 and 8.5.

### 3.3. Analysis of SDGs 4, 5 and 8 vs. Erasmus+ and Mobility Projects (ICM)

The main results for SDGs 4, 5 and 8 are as follows.

### 3.3.1. SDG 4—Ensure Inclusive and Equitable Quality Education and Promote Lifelong Learning Opportunities for All

To ensure inclusive and quality education for all and promote lifelong learning is SDG 4 of which the United Nations explains that "Obtaining a quality education underpins a range of fundamental development drivers. Major progress has been made towards increasing access to education at all levels, particularly for women and girls" [18].

The United Nations also defined 10 targets and 11 indicators for SDG 4.

The targets considered that to which Erasmus+ and the projects of mobility could contribute were [18]:

- "4.3 By 2030, ensure equal access for all women and men to affordable and quality technical, vocational and tertiary education, including university;
- 4.4 By 2030, substantially increase the number of youth and adults who have relevant skills, including technical and vocational skills, for employment, decent jobs and entrepreneurship;
- 4.5 By 2030, eliminate gender disparities in education and ensure equal access to all levels of education and vocational training for the vulnerable, including persons with disabilities, indigenous peoples and children in vulnerable situations;
- 4.7 By 2030, ensure that all learners acquire the knowledge and skills needed to promote sustainable development, including, among others, through education for sustainable development and sustainable lifestyles, human rights, gender equality, promotion of a culture of peace and non-violence, global citizenship and appreciation of cultural diversity and of culture's contribution to sustainable development".

The relevant means of implementation were [18]:

- "4.b By 2020, substantially expand globally the number of scholarships available to developing countries, in particular least developed countries, small island developing States and African countries, for enrolment in higher education, including vocational training and information and communications technology, technical, engineering and scientific programmes, in developed countries and other developing countries;
- 4.c By 2030, substantially increase the supply of qualified teachers, including through international cooperation for teacher training in developing countries, especially least developed countries and small island developing States".

In a closer look, although goals 4.3 and 4.4 are covered individually, it is important to note that they are connected.

Target 4.3: Equal access to affordable technical, vocational and higher education

Under Target 4.3—"By 2030, ensure equal access for all women and men to affordable and quality technical, vocational and tertiary education, including university"—access to higher education is frequently insufficient, particularly in least developed countries (LDCs), resulting in a knowledge gap with major social and economic consequences. Therefore, it is very important to remove obstacles to skill development and technical and vocational education and training (TVET) beginning at the secondary level as well as in tertiary education, including university.

It is also important to provide young people and adults with lifelong learning opportunities. The programme has a specific objective in the field of education and training that intends to "promote the emergence and raise awareness of a European lifelong learning area designed to complement policy reforms at national level and to support the modernisation of education and training systems, in particular through enhanced policy cooperation, better use of EU transparency and recognition tools and the dissemination of good practices" [16], which is aligned with the possible contribution to this target.

Tertiary education and universities have an essential role in promoting critical and creative thinking as well as developing and disseminating information for social, cultural, ecological and economic growth, in addition to teaching job skills, and in the creation of information and the development of analytical and creative capacities that enable solutions to be found for local and global challenges in all domains of sustainable development through their research function. They are essential for the education of future scientists, professionals and leaders, as well. An aim of mobility projects is to "support the professional development of those who work in education, training and youth with a view to innovating and improving the quality of teaching, training and youth work across Europe" [16], which is aligned with what has been previously mentioned.

An important expected outcome that arises from mobility projects for youth workers and professionals involved in education, training and youth is "improved competences,

linked to their professional profiles (teaching, training, youth work, etc.)" [16], also corroborating the previous idea. Another trend is increased staff and learner mobility, as well as a flow of students travelling abroad to further their academic qualifications.

As a result, qualification comparability, recognition and quality assurance have become rising topics of concern, particularly in countries with weak administrative institutions. At the same time, tertiary education mobility is a benefit and an opportunity that should be capitalized on in order to improve students' competences and global competitiveness. The alignment of these topics with the aims of mobility projects for higher education (ICM included) with the expected outcomes for students, trainees, apprentices, young people, youth workers and professionals involved in education, training and youth is a reality.

The following specific objective pursued by the Erasmus+ Programme in the field of education and training is also a contribution to SDG 4: to improve the level of key competences and skills, with particular regard to their relevance for the labour market and their contribution to a cohesive society, in particular through increased opportunities for learning mobility and through strengthened cooperation between the world of education and training and the world of work. Lifelong learning includes TVET and higher education (including universities), adult learning, education and training.

Promoting lifelong learning demands a sector-wide approach that includes formal, non-formal and informal learning opportunities for people of all ages as well as adult learning, education and training. It is critical to create chances for older people to have equal access to higher education, with a focus on vulnerable and/or disadvantaged groups. Erasmus+ is a tool that can provide those opportunities.

Target 4.4: Increase the number of people with relevant skills for financial success

In the context of constantly changing labour markets, rising unemployment, particularly among young people, ageing labour forces in some countries, migration and technological improvements, all countries must increase people's knowledge, skills and competences for decent work, entrepreneurship and life.

Target 4.4 envisages to "by 2030, substantially increase the number of youth and adults who have relevant skills, including technical and vocational skills, for employment, decent work and entrepreneurship" [18].

Education and training programmes in many nations are also expected to satisfy the quickly changing demands of young people and adults to upgrade and learn new skills. It is critical to expand and diversify learning opportunities through a variety of education and training modalities so that all young people and adults, particularly women and girls, can gain essential information, skills and competences for decent employment and life.

The Erasmus+ Programme and mobility projects contribute by supporting "learners in the acquisition of learning outcomes (knowledge, skills and competences) with a view to improving their personal development, their involvement as considerate and active citizens in society and their employability in the European labour market and beyond" [16]; and improving "the level of key competences and skills, with particular regard to their relevance for the labour market and their contribution to a cohesive society, in particular through increased opportunities for learning mobility and through strengthened cooperation between the world of education and training and the world of work" [16]; with the expected outcome of "greater understanding of interconnections between formal and non-formal education, vocational training and the labour market respectively" [16].

An important strategy is based on encouraging learners to take flexible learning paths in both formal and informal settings; allowing them to accumulate and transfer credits for different levels of achievement; recognizing, validating and accrediting prior learning and creating appropriate bridging programmes and career guidance and counselling services. This strategy fits in with one of the main features of the Erasmus+ Programme, namely "Recognition and validation of skills and qualifications" [16], and with one of the aims of mobility projects, such as to "reinforce synergies and transitions between formal, non-formal education, vocational training, employment and entrepreneurship; and ensure a better recognition of competences gained through the learning periods abroad" [16].

Target 4.5: Eliminate all discrimination in education

Under Target 4.5—"By 2030, eliminate gender disparities in education and ensure equal access to all levels of education and vocational training for the vulnerable, including persons with disabilities, indigenous peoples and children in vulnerable situations" [18]—poverty must remain a top focus, as it remains the single biggest barrier to inclusion at all levels and in all parts of the world.

Many education programmes have found innovative ways to aid families and learners in overcoming financial obstacles to education, and many education programmes have found innovative ways to assist families and learners in overcoming financial difficulties to education. Such approaches must be developed and scaled up. From a strategic point of view, it is critical to ensure that education policies, sector plans and budgeting uphold the principles of non-discrimination and equality in and through education, as well as to develop and implement targeted urgent strategies for vulnerable and excluded groups, as well as to develop indicators to track progress toward equality.

It is also critical to ensure that government plans, budgets, curricula and textbooks, as well as teacher training and supervision, are free of gender stereotypes and promote equality, non-discrimination and human rights, as well as intercultural education. This statement is corroborated by the Erasmus+ Programme, namely in the feature "Equity and inclusion" [16]. Being aware of linguistic and cultural diversity is a way to promote inclusion and understanding for non-discrimination, and mobility projects support that.

Education that aids in the development of peaceful and sustainable communities is critical in a worldwide world with unsolved social, political, economic and environmental concerns. However, such revolutionary ideas are rarely fully integrated into educational systems.

Target 4.7: Education for sustainable development and global citizenship

To "Ensure that all learners acquire knowledge and skills needed to promote sustainable development, including, among others, through education for sustainable development and sustainable lifestyles, human rights, gender equality, promotion of a culture of peace and non-violence, global citizenship and appreciation of cultural diversity and of culture's contribution to sustainable development" [18] by 2030 is Target 4.7 for SDG 4. Under this target, it is also critical to prioritize education's role in the realization of human rights, peace and responsible citizenship from local to global levels, gender equality, sustainable development and health in SDG 4-Education 2030.

One promising strategy is to provide learners of all ages and genders with opportunities to acquire the knowledge, skills, values and attitudes necessary to develop peaceful, healthy and sustainable societies throughout their lives. The Erasmus+ Programme contributes to the achievement of the promotion of European values in accordance with Article 2 of the Treaty on the European Union, namely "The Union is founded on the values of respect for human dignity, freedom, democracy, equality, the rule of law and respect for human rights, including the rights of persons belonging to minorities. These values are common to the Member States in a society in which pluralism, non-discrimination, tolerance, justice, solidarity and equality between women and men prevail" [19], which corroborates Target 4.7.

Target 4.b: Expand higher education scholarships for developing countries

Concerning the means of implementation—"Target 4.b: By 2020, substantially expand globally the number of scholarships available to developing countries, in particular least developed countries, small island developing States and African countries, for enrolment in higher education, including vocational training and information and communications technology, technical, engineering and scientific programmes, in developed countries and other developing countries" [18]—scholarship programmes can be quite beneficial in giving chances to young people and adults who might otherwise be unable to continue their education. They also make a significant contribution to the internationalization of tertiary education and research systems, especially in LDCs.

They can aid in expanding access to global information and increasing capability for transferring and adapting knowledge and technology to local contexts. Scholarships

should be publicly aimed at young people from disadvantaged backgrounds, in keeping with the SDG 4-Education 2030 focus on equity, inclusion and quality. The strategy to develop joint programmes between universities in the home country and the recipient country to encourage students to return home, as well as other mechanisms to prevent brain drain—the emigration of highly trained people—and promote brain gain fits the mobility project goals and is aligned with the Erasmus+ Programme's specific goal.

"Enhance the international dimension of education and training, in particular through cooperation between Programme and Partner-Country institutions in the field of VET and in higher education, by increasing the attractiveness of European higher education institutions and supporting the EU's external action, including its development objectives, through the promotion of mobility and cooperation between Programme and Partner-Country higher education institutions and targeted capacity building in Partner Countries" [16]. Mobility periods normally create the conditions for participants to return to home countries, promoting brain gain and not brain drain.

Target 4.c: Increase the supply of qualified teachers in developing countries

"Target 4.c: By 2030, substantially increase the supply of qualified teachers, including through international cooperation for teacher training in developing countries, especially least developed countries and small island developing States" [18] suggests that teachers and educators must be empowered, appropriately recruited and compensated, motivated, properly qualified and supported within well-resourced, efficient and well-managed systems in order to ensure quality education.

Professional development that supports teachers' personal learning and advancement throughout their careers has been a priority of successful education systems that assure quality and equity. Teachers, with the help of school administrators, government officials and communities, make a significant contribution to improving student learning outcomes.

Teachers are receptive to change and want to learn and progress throughout their careers, according to data. At the same time, they require more time and space to take more initiatives in collaboration with colleagues and school leaders as well as to take advantage of professional development opportunities.

The Erasmus+ Programme encourages professional development through new experiences and the establishment of a new network of contacts through mobility programmes, contributing to the achievement of this target.

### 3.3.2. SGD 5—Achieve Gender Equality and Empower all Women and Girls

SDG 5 concerns gender equality, and according to the UN explanation [19], "Gender equality is not only a fundamental human right, but a necessary foundation for a peaceful, prosperous and sustainable world. There has been progress over the last decades: More girls are going to school, fewer girls are forced into early marriage, more women are serving in parliament and positions of leadership, and laws are being reformed to advance gender equality" [20]. "Providing women and girls with equal access to education, health care, decent work, and representation in political and economic decision-making processes will fuel sustainable economies and benefit societies and humanity at large" [21]. The United Nations have defined 9 targets and 14 indicators for SDG 5. Erasmus+, through its contribution to promoting European values in accordance with Article 2 of the Treaty on the European Union, is clearly contributing to this objective, since among the values it defends are human dignity and equality.

Target 5.1: End discrimination against women and girls

Target 5.c: Adopt and strengthen policies and enforceable legislation for gender equality

The objective, "5.1 End all forms of discrimination against all women and girls everywhere", and the means of implementation, "5.c Adopt and strengthen sound policies and enforceable legislation for the promotion of gender equality and the empowerment of all women and girls at all levels", benefit from the contribution of Erasmus+ and mobility projects.

To reinforce this statement, in addition to what was mentioned above, we also have one of the objectives of the cooperation between the EU-eligible Partner Countries, to which this action contributes to "promote the development and external policy objectives and principles including national ownership, social cohesion, equity, proper geographical balance and diversity. Special attention will be given to the least developed countries as well as to disadvantaged students from poor socio-economic backgrounds and to students will special needs" [21]. It also contributes to the expected results, namely "better awareness of the European project and the EU values" [16] and "increased ability to address the needs of the disadvantaged" [21].

### 3.3.3. SDG 8—Promote Sustained, Inclusive and Sustainable Economic Growth, Full and Productive Employment and Decent Work for All

To promote inclusive and sustainable economic growth, employment and decent work for all is Sustainable Development Goal 8, which has 12 targets and 17 indicators.

According to the United Nations, "Sustained and inclusive economic growth can drive progress, create decent jobs for all and improve living standards" [22]. "Roughly half the world's population still lives on the equivalent of about US$2 a day. Moreover, in too many places, having a job does not guarantee the ability to escape from poverty. This slow and uneven progress requires us to rethink and retool our economic and social policies aimed at eradicating poverty" [23].

Target 8.3: Promote policies to support job creation and growing enterprises
Target 8.5: Full employment and decent work with equal pay
Target 8.7: End modern slavery, trafficking and child labour

The objectives "8.3 Promote development-oriented policies that support productive activities, decent job creation, entrepreneurship, creativity and innovation, and encourage the formalization and growth of micro-, small- and medium-sized enterprises, including through access to financial services" [23], "8.5 By 2030, achieve full and productive employment and decent work for all women and men, including for young people and persons with disabilities, and equal pay for work of equal value" [23] and "8.6 By 2020, substantially reduce the proportion of youth not in employment, education or training" [23] were selected.

Having analysed these objectives and the respective indicators, we verify that the Erasmus+ Programme and the mobility projects also contribute to this objective, since there is a specific objective pursued by the Erasmus+ Programme in the field of education and training that aims to "improve the level of key competences and skills, with particular regard to their relevance for the labour market and their contribution to a cohesive society, in particular through increased opportunities for learning mobility and through strengthened cooperation between the world of education and training and the world of work" and two objectives in mobility projects that "support the professional development of those who work in education, training and youth with a view to innovating and improving the quality of teaching, training and youth work across Europe" [23] and "reinforce synergies and transitions between formal, non-formal education, vocational training, employment and entrepreneurship; and, ensure a better recognition of competences gained through the learning periods abroad" [23].

For youth workers and professionals involved in education, training and youth, the mobility activities are expected to produce outcomes such as "better quality of their work and activities in favour of students, trainees, apprentices, pupils, adult learners, young people and volunteers; increased opportunities for professional and career development; and, increased motivation and satisfaction in their daily work" [23], which also corroborates the relationship between the programme, the mobility projects and SDG 8.

## 4. Final Considerations

Mobility projects, such as the ICM, are very relevant within the Erasmus+ Programme. Although a selection of goals and targets was made separately for the Programme and for

the mobility projects, the analysis of the results was made in a holistic way covering both dimensions. We believe it is important to mention at this stage that Erasmus+ has other key actions that cover other aspects, such as cooperation for innovation and exchange of good practices and support for policy reform, among others, which can be equally important contributions to several SDGs.

Analysing the present information on the Erasmus+ Programme, especially in the fields of education and training and mobility projects in higher education, and considering Sustainable Development Goals 4, 5 and 8 and the associated indicators, it was possible to respond to several of them with a positive correspondence through this opportunity offered by the European Commission. The objectives and means of implementation with a positive contribution by the Erasmus+ and mobility projects were SDG 4: 4.3, 4.4, 4.5, 4.7, 4.b and 4.c; SDG 5: 5.1 and 5.c and SDG 8: 8.3 and 8.5.

According to [24], there are several benefits from an academic mobility experience that can be expressed in a form of key criteria, such as participants in student exchange programmes, and can improve their employability on a personal level as well as their self-sufficiency and train their intercultural skills: academic exchange programmes also serve as a platform for learning new and improving current foreign language skills and allow universities to share best practices and make the learning process more transparent. Student and staff mobility is one of the most important tools for achieving the SDGs through internationalization. Together, they can help to improve educational quality (SDG 4). Mobility has the potential to shape and enhance vast and shared regional identities, resulting in the development of a unified sense of global citizenship [25]. Academic mobility, both inbound and outbound, can aid in the creation of direct and indirect cross-sector partnerships. Governments can help HEIs attract international students and encourage local students to go to other countries to learn deeply and more about how other cultures/societies operate [26]. Ref. [27], through their study, eventually demonstrated that despite the small number of Erasmus+ students present, they were able to present a different approach to the daily problems of a farm, using terms and concepts of a responsible approach to production. From what has just been written, we can see that mobility in the context of higher education contributes to goals that integrate SDGs, such as SDG 4 and SDG 8, and also, indirectly, to SDG 5, through awareness of other cultures and societies. These are examples that confirm the practical effects of the Erasmus+ Programme in higher education and, through this, the contribution to SDGs.

Although the contents of SDGs were analysed individually, it was noticed that there are relationships with variable degrees of strength between them. However, in this work, we performed an individual analysis of each SDG and presented those that seemed more consistent in relation to the contributions that the Erasmus+ Programme could give to its effective implementation. Another limitation to this work is the impossibility of analysing the content of all mobility projects submitted by HEIs in the period 2014–2020 under the International Credit Mobility, where their internationalisation strategies with Partner Countries are defined and which could make further contributions to sustainable development under other SDGs.

As future research proposals we suggest analysis of SDGs 9, 10, 16 and 17 within the reality of the Erasmus+ Programme and International Credit Mobility projects, analysis of the relationship between the SDGs to which the Erasmus+ Programme and ICM Projects have contributed and analysis of the SDGs associated with other key actions of the Erasmus+ Programme, such as capacity building projects in the field of higher education, the objective of which being to support the modernisation, accessibility and internationalisation of higher education in Partner Countries.

**Author Contributions:** Conceptualization, T.N. and M.S.; methodology, T.N.; validation, T.N., M.S., F.J. and E.C.; formal analysis, T.N.; investigation, T.N. and M.S.; resources, T.N.; data curation, T.N. and M.S.; writing—original draft preparation, T.N and M.S.; writing—review and editing, T.N. and M.S.; visualization, F.J., E.C. and M.S.; supervision, E.C.; project administration, T.N., M.S., F.J. and E.C.; funding acquisition, M.S. and E.C. All authors have read and agreed to the published version of the manuscript.

**Funding:** This research was funded by Fundação para a Ciência e a Tecnologia, grant number PTDC/CED-EDG/29252/2017.

**Institutional Review Board Statement:** Not applicable.

**Informed Consent Statement:** Not applicable.

**Data Availability Statement:** All documents, data and information used in this work are available to the public.

**Conflicts of Interest:** The authors declare no conflict of interest.

**Appendix A**

The meaning of each selected objective is as follows in Table A1. For each of these objectives/targets there are associated indicators which support the analysis of the information and which are set out below:

**Table A1.** Meaning of selected objectives.

| Target Number | Selected SDG Target and Mean of Implementation | Indicator(s) |
|---|---|---|
| 4.3 | By 2030, ensure equal access for all women and men to affordable and quality technical, vocational and tertiary education, including university | 4.3.1 Participation rate of youth and adults in formal and non-formal education and training in the previous 12 months, by sex |
| 4.4 | By 2030, substantially increase the number of youth and adults who have relevant skills, including technical and vocational skills, for employment, decent jobs and entrepreneurship | 4.4.1 Proportion of youth and adults with information and communications technology (ICT) skills, by type of skill |
| 4.5 | By 2030, eliminate gender disparities in education and ensure equal access to all levels of education and vocational training for the vulnerable, including persons with disabilities, indigenous peoples and children in vulnerable situations | 4.5.1 Parity indices (female/male, rural/urban, bottom/top wealth quintile and others such as disability status, indigenous peoples and conflict-affected, as data become available) for all education indicators on this list that can be disaggregated |
| 4.7 | By 2030, ensure that all learners acquire the knowledge and skills needed to promote sustainable development, including through, among others, education for sustainable development and sustainable lifestyles, human rights, gender equality, promotion of a culture of peace and non-violence, global citizenship and appreciation of cultural diversity and of culture's contribution to sustainable development | 4.7.1 Extent to which (i) global citizenship education and (ii) education for sustainable development are mainstreamed in (a) national education policies, (b) curricula, (c) teacher education and (d) student assessment |

**Table A1.** *Cont.*

| Target Number | Selected SDG Target and Mean of Implementation | Indicator(s) |
|---|---|---|
| **4.b** | By 2020, substantially expand globally the number of scholarships available to developing countries, in particular least developed countries, small island developing states and African countries, for enrolment in higher education, including vocational training and information and communications technology, technical, engineering and scientific programmes, in developed countries and other developing countries | 4.b.1 Volume of official development assistance flows for scholarships by sector and type of study |
| **5.1** | End all forms of discrimination against all women and girls everywhere | 5.1.1 Whether or not legal frameworks are in place to promote, enforce and monitor equality and non-discrimination on the basis of sex |
| **5.c** | Adopt and strengthen sound policies and enforceable legislation for the promotion of gender equality and the empowerment of all women and girls at all levels | 5.c.1 Proportion of countries with systems to track and make public allocations for gender equality and women's empowerment |
| **8.3** | Promote development-oriented policies that support productive activities, decent job creation, entrepreneurship, creativity and innovation and encourage the formalization and growth of micro-, small- and medium-sized enterprises, including through access to financial services | 8.3.1 Proportion of informal employment in total employment, by sector and sex |
| **8.5** | By 2030, achieve full and productive employment and decent work for all women and men, including for young people and persons with disabilities, and equal pay for work of equal value | 8.5.1 Average hourly earnings of employees, by sex, age, occupation and persons with disabilities |
| **8.7** | Take immediate and effective measures to eradicate forced labour, end modern slavery and human trafficking and secure the prohibition and elimination of the worst forms of child labour, including recruitment and use of child soldiers, and by 2025 end child labour in all its forms | 8.7.1 Proportion and number of children aged 5–17 years engaged in child labour, by sex and age |

Source: Our own elaboration supported by global indicator framework adopted by [28].

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
