# Peer review of "The Erasmus+ Programme and Sustainable Development Goals—Contribution of Mobility Actions in Higher Education"

_sustainability, doi:10.3390/su14031628_

Round 1
Reviewer 1 Report
The research question "Which SDGs do Erasmus+ and higher education mobility projects directly contribute to?" has been answered throughout the document in an excellent way.
It would be excellent if you could complete the study with a detailed analysis of SDGs 9, 10, 16 and 17 within the Erasmus+ program, as well as the relationship between the different objectives and their goals.
Author Response
Thank you for your comments. The work will continue.
Reviewer 2 Report
In my opinion, based on the review of information on Sustainable Development Goals (SDG) and the Erasmus+ program, it would be possible to formulate a research problem that the authors wanted to solve in their research presented in the article. The need to fill the knowledge gap on the borderline between SDG and Erasmus+ can also be mentioned. This gap filling would be related to the question formulated by the Authors in lines 160-162.
I propose that the article should clearly state the purpose(s) of the research. I suggest writing the following sentence at the end of the Introduction: "The purpose of the research was ...". Moreover, when formulating the research goal(s), it would be worth indicating what was the cognitive (scientific) goal and what was the utilitarian (useful) goal.
In the Abstract, the authors used the acronym SDG twice, but never gave the full name of this acronym. A reader who has access only to the Abstract may find from the title of the article that the SDG can be associated with the Sustainable Development Goals. However, in my opinion, the reader should be sure of certain information, and not necessarily guess at it. Although the full name of the acronym SDG was given in the body of the article, the Abstract is an equally important part of the article.
If the Authors asked the main question in the research (in lines: 160-162), why does this question not end with a question mark? I think that for the correctness of the statements used, it is also worth using appropriate characters (e.g. question marks). The same remark applies to the sentence formulated in lines: 187-188. If a parenthesis was opened on line 165, where did it close?
If, in the "Methods" chapter, the Authors asked a research question, and previously mentioned issues related to the goals set in the Erasmus + program, it would be worthwhile to clearly state what the criteria of comparison with the goals set in the SDG were adopted. In general, it is mentioned in the Methods chapter, but - in my opinion - when formulating an approach to research, it is necessary to clearly indicate the criteria for comparisons, using the word "criteria".
In my opinion, the Methods section should be entitled Materials and Methods. Comparing the goals of the SDG and the assumptions / goals of the Erasmus + program, the Authors used specific materials, which were to some extent presented in the article and listed in References, hence my suggestion regarding the title of the chapter.
In chapter 2 (Methods), the authors wrote that "... the team analyzed the targets, the means of implementation and the respective indicators for each of the 17 SDGs". I would like to ask about the "indicators" mentioned in this sentence. What specific indicators were analysed? What were the names of these indicators? What was their interpretation?
I think that in the presented analysis it would be worth referring to the issues included in the Staff Mobility for Teaching - Mobility Agreement, which is filled in by academic teachers applying for a trip to a foreign university under the Erasmus+ program. This document covers the following issues that need to be planned:
- Overall objectives of the mobility;
- Added value of the mobility (in the context of the modernization and internationalization strategies of the institutions involved);
- Content of the teaching program;
- Expected outcomes and impact (e.g. on the professional development of the teaching staff member and on the competences of students at both institutions).
The issues presented have been generally included in the chapter on Methods, but perhaps some elements could be supplemented.
I would like to ask what is / what would be the practical effect of comparing the goals set in the SDG and in the implementation of the Erasmus+ program? What does this result for the practice or what recommendations could the authors formulate for, for example, the Erasmus+ program as a result of a comparison with the goals of the SDG? I think that the article could provide some practical examples of the effects of the Erasmus+ program to show its potential. An example of the effects of implementing the Erasmus+ program with students is presented in the article "The topic of the ideal dairy farm can inspire how to assess knowledge about dairy production processes: A case study with students and their contributions". This example shows that the Erasmus+ program and the teaching activities implemented under the program can inspire a new approach to the assessment of learning outcomes. Perhaps the authors could also cite other publications confirming the practical effects of the Erasmus+ program in higher education.
Author Response
Here is the answer to reviewer 1 comments:
In my opinion, based on the review of information on Sustainable Development Goals (SDG) and the Erasmus+ program, it would be possible to formulate a research problem that the authors wanted to solve in their research presented in the article. The need to fill the knowledge gap on the borderline between SDG and Erasmus+ can also be mentioned. This gap filling would be related to the question formulated by the Authors in lines 160-162.
I propose that the article should clearly state the purpose(s) of the research. I suggest writing the following sentence at the end of the Introduction: "The purpose of the research was ...". Moreover, when formulating the research goal(s), it would be worth indicating what was the cognitive (scientific) goal and what was the utilitarian (useful) goal.
Answer: Corrected accordingly to what has been proposed by the reviewer.
In the Abstract, the authors used the acronym SDG twice, but never gave the full name of this acronym. A reader who has access only to the Abstract may find from the title of the article that the SDG can be associated with the Sustainable Development Goals. However, in my opinion, the reader should be sure of certain information, and not necessarily guess at it. Although the full name of the acronym SDG was given in the body of the article, the Abstract is an equally important part of the article.
Answer: Corrected accordingly to reviewer's suggestion.
If the Authors asked the main question in the research (in lines: 160-162), why does this question not end with a question mark? I think that for the correctness of the statements used, it is also worth using appropriate characters (e.g. question marks). The same remark applies to the sentence formulated in lines: 187-188. If a parenthesis was opened on line 165, where did it close?
Answer: Corrected accordingly to reviewer's indicaton.
If, in the "Methods" chapter, the Authors asked a research question, and previously mentioned issues related to the goals set in the Erasmus + program, it would be worthwhile to clearly state what the criteria of comparison with the goals set in the SDG were adopted. In general, it is mentioned in the Methods chapter, but - in my opinion - when formulating an approach to research, it is necessary to clearly indicate the criteria for comparisons, using the word "criteria".
In my opinion, the Methods section should be entitled Materials and Methods. Comparing the goals of the SDG and the assumptions / goals of the Erasmus + program, the Authors used specific materials, which were to some extent presented in the article and listed in References, hence my suggestion regarding the title of the chapter.
Answer: Corrected accordingly to reviewer's suggestion.
In chapter 2 (Methods), the authors wrote that "... the team analyzed the targets, the means of implementation and the respective indicators for each of the 17 SDGs". I would like to ask about the "indicators" mentioned in this sentence. What specific indicators were analysed? What were the names of these indicators? What was their interpretation?
Answer: Corrected accordingly to reviewer's suggestion.
I think that in the presented analysis it would be worth referring to the issues included in the Staff Mobility for Teaching - Mobility Agreement, which is filled in by academic teachers applying for a trip to a foreign university under the Erasmus+ program. This document covers the following issues that need to be planned:
- Overall objectives of the mobility;
- Added value of the mobility (in the context of the modernization and internationalization strategies of the institutions involved);
- Content of the teaching program;
- Expected outcomes and impact (e.g. on the professional development of the teaching staff member and on the competences of students at both institutions).
The issues presented have been generally included in the chapter on Methods, but perhaps some elements could be supplemented.
Answer: The outcomes analysed were those mentioned in the regulation as expected in a general way and not those effectively obtained through the analysis of questionnaires of participants in mobilities under Erasmus+.
I would like to ask what is / what would be the practical effect of comparing the goals set in the SDG and in the implementation of the Erasmus+ program? What does this result for the practice or what recommendations could the authors formulate for, for example, the Erasmus+ program as a result of a comparison with the goals of the SDG? I think that the article could provide some practical examples of the effects of the Erasmus+ program to show its potential. An example of the effects of implementing the Erasmus+ program with students is presented in the article "The topic of the ideal dairy farm can inspire how to assess knowledge about dairy production processes: A case study with students and their contributions". This example shows that the Erasmus+ program and the teaching activities implemented under the program can inspire a new approach to the assessment of learning outcomes. Perhaps the authors could also cite other publications confirming the practical effects of the Erasmus+ program in higher education.
Answer: That information was added in the final point of Final Considerations.